# Cognitive Function Improvement in Mouse Model of Alzheimer’s Disease Following Transcranial Direct Current Stimulation

**DOI:** 10.3390/brainsci10080547

**Published:** 2020-08-12

**Authors:** Wang-In Kim, Jae-Young Han, Min-Keun Song, Hyeng-Kyu Park, Jihoon Jo

**Affiliations:** 1Department of Physical & Rehabilitation Medicine, Regional Cardiocerebrovascular Center, Center for Aging and Geriatrics, Chonnam National University Medical School & Hospital, Gwangju 61469, Korea; wangto9@naver.com (W.-I.K.); drsongmk@daum.net (M.-K.S.); phk1118@naver.com (H.-K.P.); 2NeuroMedical Convergence Lab, Biomedical Research Institute, Chonnam National University Hospital, Jebong-ro, Gwangju 61469, Korea; Jihoon.Jo@jnu.ac.kr; 3Department of Biomedical Sciences, BK21 PLUS Center for Creative Biomedical Scientists at Chonnam National University, Research Institute of Medical Sciences, Chonnam National University Medical School, Gwangju 61469, Korea; 4Department of Neurology, Chonnam National University Medical School, Gwangju 61469, Korea

**Keywords:** Alzheimer’s disease, direct current stimulation, cognitive enhancement, mouse model

## Abstract

Anodal transcranial direct current stimulation (tDCS) is a painless noninvasive method that reportedly improves cognitive function in Alzheimer’s disease (AD) by stimulating the brain. However, its underlying mechanism remains unclear. Thus, the present study investigates the cognitive effects in a 5xFAD AD mouse model using electrophysiological and pathological methods. We used male 5xFAD C57BL/6J and male C57BL/6J wild-type mice; the dementia model was confirmed through DNA sequencing. The verified AD and wild-type mice were randomly assigned into four groups of five mice each: an induced AD group receiving tDCS treatment (Stim-AD), an induced AD group not receiving tDCS (noStim-AD), a non-induction group receiving tDCS (Stim-WT), and a non-induction group not receiving tDCS (noStim-WT). In the Stim group, mice received tDCS in the frontal bregma areas at an intensity of 200 µA for 20 min. After 2 weeks of treatment, we decapitated the mice, removed the hippocampus from the brain, confirmed its neuronal activation through excitatory postsynaptic potential (EPSP) recording, and performed molecular experiments on the remaining tissue using western blots. EPSP significantly increased in the Stim-AD group compared to that in the noStim-AD, which was comparable to that in the non-induced groups, Stim-WT and noStim-WT. There were no significant differences in cyclic amp-response element binding protein (CREB), phosphorylated CREB (pCREB), and Brain-derived neurotrophic factor (BDNF) levels in the Stim-AD group compared to those in the noStim-AD group. This study demonstrated that a tDCS in both frontal lobes of a transgenic 5xFAD mouse model affects long-term potentiation, indicating possible enhancement of cognitive function.

## 1. Introduction

Alzheimer’s disease (AD), with anterograde amnesia as its key symptom, is a neurodegenerative disorder characterized by brain dysfunction including the loss of cognitive function and behavior. AD dementia is particularly prevalent in the elderly population and deteriorates the quality of life in the aging population [1]. According to reported data, medical treatments for dementia include cholinesterase inhibitors and N-methyl-D-aspartate (NMDA) receptor antagonists [2].

However, these drugs are ineffective in some patients, are expensive, and are prone to side effects after long-term use. Thus, alternative or supplementary therapy is drawing significant attention [3,4].

Since direct stimulation is clinically ineffective, the indirect method of noninvasive transcranial magnetic/electrical stimulation is being actively researched. This stimulation method has been applied in various fields and proven effective according to clinical and animal research. Transcranial direct current stimulation (tDCS) is a brain stimulation technique used to neuromodulate a brain area through small electrodes that emit a weak direct current on the skull [5].

Clinical studies have reported that tDCS improves cognitive function in patients of stroke, Parkinson’s disease, and AD with cognitive impairment [6,7,8,9,10]. Since the brain tissue cannot be directly examined, it is difficult in clinical research to verify the presence of cognitive enhancement and its exact mechanism [11,12]. Therefore, uncovering the exact mechanism of tDCS cannot be achieved through clinical research alone. To overcome such limitations, animal models are being used and many studies have shed light on the mechanism behind the cognitive enhancement effect [13,14,15,16,17].

Nevertheless, most animal studies of dementia involve artificial models that artificially induce cognitive deficits or cause acute neurological problems that damage brain function. To complement this limitation, genetic models similar to the induced disease model have been generated, as well as genetic animal models of dementia [18,19].

In the present study, we aimed to identify the effects of tDCS on cognitive function improvement and the underlying mechanism using the genetic dementia model 5xFAD.

## 2. Materials and Methods

We crossed wild-type (C57BL/6J (Damul Science in Korea)) female mice and 5X five familial AD (FAD) amyloid beta precursor protein (APP) KM 670/671NL (Sweden)) male mice to obtain a mixed strain and selected aged mice with cognitive deficits. Three weeks post birth, we collected the ear tissue of the offspring for genotyping and used only male mice confirmed as 5xFAD or wild type. After confirming the mutation, we experimented with mice that were 4 months old. Because 5xFAD mice were violent, they were individually housed, with food ad libitum under a 12 h:12 h light–dark cycle at a temperature of 23–30 °C. All animal experiments were approved by the Institutional Animal Care and Use Committee of Chonnam National University (CNU IACUC-H-2018-53).

### 2.1. The tDCS Treatment

For tDCS stimulation, we used a battery-driven, constant-current stimulator (HDCprog manufactured by Newronika s.r.l., Italy and distributed by Magstim Co. Ltd., UK). For the two-channel anodal method, we positioned cup-shaped active electrodes (1 cm × 1 cm) on the frontal skull of both hemispheres, whereas for the cathodal method, a 0.5-cm sponge pad was placed on the neck. Electrical stimulation was performed at an intensity of 0.2 mA for 20 min over a period of 2 weeks (5 consecutive days followed by a 2-day break). Animals were randomly allocated into four groups (five animals per group), with the Stim-AD group 5xFAD mice receiving tDCS treatment, the noStim-AD 5xFAD mice receiving no tDCS, the Stim-WT wild-type mice receiving tDCS, and the noStim-WT group wild-type mice receiving no tDCS. The noStim-AD and noStim-WT groups were left in the cage for 20 min without the stimulation pads on. Stim-AD and Stim-WT mice had two small circular pads on the head.

### 2.2. Brain Tissue Preparation

We sacrificed mice 2 weeks post treatment and immediately extracted the brains. Brains were immersed in Artificial cerebrospinal fluid (ACSF) (125 mM NaCl, 2.8 mM KCl, 26 mM NaHCO_3_, 1.25 mM NaH_2_PO_4_, 2 mM CaCl_2_, 1 mM MgSO_4_) and 10 mM freezing liquid, and were then placed on a cooling pad to dissect the hippocampus. Hippocampal slices of Stim-AD, noStim-AD, Stim-WT, and noStim-WT mice were prepared using the rotary slicer and automatic chopper to generate 400-µm thick samples (Mickle Laboratory Engineering Co. Ltd.). The slices were stabilized for at least an hour in ACSF and gassed with 95% O_2_ and 5% CO_2_.

### 2.3. Excitatory Postsynaptic Potential (EPSP) Recording

Hippocampal slices were transferred to a glass-bottomed submersion recording chamber and continuously perfused with ACSF (95% O_2_/5% CO_2_) at room temperature. We used bipolar nichrome electrodes for each slice recording and an isolation unit for the stimulator. For the recording electrode, we used a glass micropipette filled with 3M NaCl, mounted in an electrode holder. We recorded the field excitatory postsynaptic potential (fEPSP) in the stratum corneum of the CA1 region while delivering 10-µs pulses through a bipolar tungsten electrode at 15-s intervals. To assess synaptic transmission, we used the initial slope of the fEPSP. We adjusted the baseline to the value that elicited a response equal to about 30% of the maximal response. For Long-term potentiation (LTP) induction, we applied high-frequency stimulation (100 pulses at 100 Hz) in the Schaffer collateral. We used the stimulation strength that elicited a stable baseline for 30 min in a follow-up recording and then recorded LTP for 60 min. Data were analyzed using the WinLTP software.

### 2.4. Western Blot

We prepared hippocampal slices from the remaining brain tissue after recording EPSP. We homogenized the samples on ice using a Radioimmunoprecipitation assay (RIPA) buffer and centrifuged them at 4 °C for 30 min at 13,000 rpm. We used a Bicinchonic acid (BCA) assay for protein quantification. We loaded the quantified protein onto separate 10–12% gels and transferred it to a Polyvinylidene fluoride (PVDF) membrane. We first incubated the membrane in 5% non-fat milk for 1 h at room temperature for blocking. The membrane was then rinsed three times in tris buffered saline buffer with tween 20 (TBST). Next, we incubated the membrane overnight at 4 °C in primary antibody solution (BDNF (1:1000), CREB (1:1000) and pCREB (1:1000)). The following morning, we rinsed the membrane three times in TBST and incubated in the secondary antibody solution (rabbit immunoglobulin G (IgG) (1:5000)) for 90 min at room temperature and rinsed three times in TBST. We incubated the membrane in the chemiluminescent horseradish peroxidase (HRP) substrate kit and detected using UVITEC Mini HD9 acquisition system (Alliance UVItec Ldt, Cambridge, UK).

### 2.5. Statistical Analysis

We performed a Kruskal–Wallis test to identify the interactions and differences between groups (noStim-AD, Stim-AD, noStim-WT, and Stim-WT) and Tukey’s test using ranks as a post hoc analysis. All data are presented as mean ± SD or mean ± SE, the statistical analysis was done using SPSS ver.25.0 (IBM, SPSS, Armonk, NY, USA), and the significance level was set to *p* < 0.05.

## 3. Results

### 3.1. tDCS Could Improve the Slope of f-EPSP

We confirmed between-group differences among the four groups. Groups other than noStim-AD (Stim-AD, Stim-WT, and noStim-WT) showed similar results. Stim-AD mice (153.2 ± 21.4%) demonstrated a more significant improvement than noStim-AD (113.9 ± 18.8%) (*p* = 0.024). The value of noStim-WT (154.2 ± 13.1%) was significantly higher than the value of noStim-AD (*p* = 0.021). There was no significance difference among the Stim-AD, Stim-WT (147.9 ± 22.2%), and noStim-WT groups (Figure 1). The result suggests that tDCS induces no difference in the WT model but does induce changes in hippocampal synaptic plasticity in the AD model.

### 3.2. Expression of tDCS Protein Level

We performed western blot analysis to identify the molecular mechanism underlying the long-term effects of tDCS on cognitive enhancement. There were no significant differences in CREB, pCREB, or BDNF protein expression in the Stim-AD group relative to the noStim-AD group. (Figure 2).

## 4. Discussion

This study investigated the effects of tDCS on the improvement of cognitive function in an Alzheimer’s dementia mouse model. After 2 weeks of treatment (five sessions per week), LTP was enhanced and neuronal activity in the hippocampus also increased, as suggested by the increase in pCREB and CREB. The possible mechanism involved is that tDCS stimulates the cortex, which in turn influences the deeply nested hippocampus.

Clinically, it has been reported that tDCS significantly affects cognitive enhancement in patients with mild Alzheimer’s dementia and improves spatial task performance and memory [20,21]. However, clinical studies are limited since brain tissues cannot be directly analyzed. We thus aimed to identify the effects and mechanism of tDCS using animal models. The 5xFAD dementia model mouse is a genetic dementia model with features resembling those of Alzheimer’s dementia (e.g., increased Aβ plaque levels, neurosis, synapse degeneration, neuronal loss, and progressive cognitive deficit) [19].

Since the hippocampus is the control tower of the brain for learning and memory and plays an important role in regulating cognitive function, hippocampal LTP is used as a proxy of basic synapse metabolism for learning and memory [22]. In the present study, we stimulated the brain using tDCS for 2 weeks, extracted the hippocampus in vivo and recorded the EPSP. The results showed that LTP was enhanced in the Stim-AD group of 5xFAD mice compared to that in the noStim-AD group, which was similar to the noStim-WT LTP results. This suggests that that the outcome of tDCS in dementia mice is hippocampal LTP. These results are consistent with previous clinical findings, where tDCS induced cortical changes or simulated the effects of LTP [23,24].

To explore the mechanism of stimulation, we examined the levels of BDNF, CREB, and pCREB, which are factors involved in cellular regeneration and proliferation in the brain. BDNF, a critical regulator of synaptic plasticity, serves as a molecular switch for LTP induction and initiation of biochemical changes [25,26]. In this study, BDNF levels showed non-significant differences in the Stim-AD group compared to those in the noStim-AD group. Since the degree of BDNF increase may change with the amount of time elapsed after stimulation, time-dependent changes must be monitored in future research [27].

CREB and pCREB, reported to improve cognitive deficits in Alzheimer’s dementia, work as a trigger for memory formation in spatial and social learning and regulate the expression of synapse- or memory-related genes [27,28,29,30]. The results of this experiment showed that CREB and pCREB showed non-significant differences in the Stim-AD group compared to those in the noStim-AD group.

Recent studies using the 3xTg model, generated by injecting an APP_swe_, PS1_M146V_, and tau- expressing P301L mutant, have reported findings contradictory to ours: tDCS had no significant effect on memory enhancement [14,19]. This may be due to the different parameters used: while Gondard et al. performed tDCS at an intensity of 50 µA for 3 weeks, in the present study, we stimulated the mice brains at 200 µA for 2 weeks while using a different transgenic model (5xFAD): the 5xFAD model generated by the co-expression of a total of five FAD mutations (APP K670N/M671L (Sweden), I716V (Florida), V717I (London), PSq M146L, and L286V) [3,19]. This resulted in a different stimulus that may be connected to tDCS parameters (e.g., the stimulation method, intensity, duration, area, and electrode size), consistent with previous findings that tDCS-induced brain activation depends on variables like electrode size, position, and conductive medium [31]. Therefore, more research is needed to find the appropriate parameters of tDCS, as the results may vary with the different parameters used.

However, this study has a limitation that a behvioral test was not conducted to confirm the models and to observe the clinical function.

## 5. Conclusions

In conclusion, this study showed that tDCS application in 5xFAD mice, a genetic animal model designed as a disease model, induces LTP response via the continuous production of pCREB, CREB, and BDNF and presented a mechanism for memory enhancement in Alzheimer’s dementia.

## Figures and Tables

**Figure 1 brainsci-10-00547-f001:**
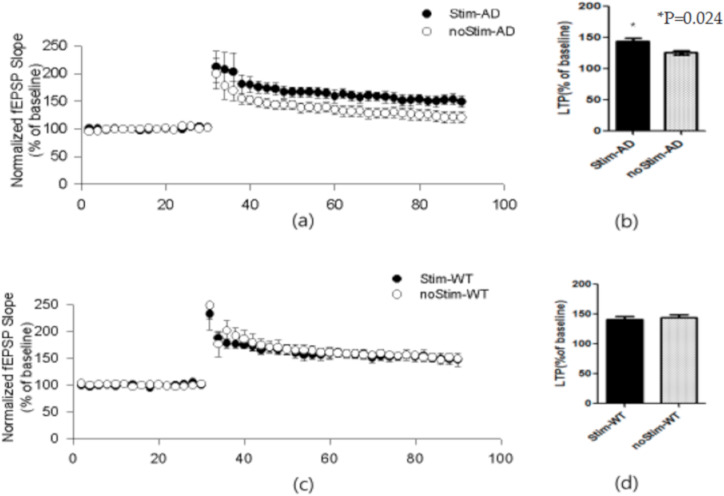
The numbers represent the field excitatory postsynaptic potential (fEPSP) response of each group. The 5xFAD induced Alzheimer’s disease (AD) mice and wild-type (WT) mice that received 2 weeks of Transcranial direct current stimulation (tDCS) showed significant changes in hippocampal Long-term potentiation (LTP) compared to those that received no stimulation. (**a**) We recorded the hippocampal LTP in 5xFAD mice that either received tDCS or no stimulation. (**b**) The bars represent the LTP amplitudes of the two AD groups of 5xFAD mice as calculated from the mean fEPSP slope over 90 min. Significance relative to noStim-AD was * *p* < 0.05. (**c**) We recorded the hippocampal LTP in WT mice that either received tDCS or no stimulation. (**d**) The bars represent the LTP amplitudes of the two WT groups as calculated from the mean fEPSP slope over 90 min. Stim-AD is a 5xFAD mouse with tDCS, while noStim-AD is a 5xFAD mouse without tDCS. Stim-WT is a WT mouse with tDCS while noStim-WT is a WT mouse without tDCS. Data are presented as mean ± SE and we performed a Kruskal–Wallis test and Tukey’s test using ranks as a post hoc analysis.

**Figure 2 brainsci-10-00547-f002:**
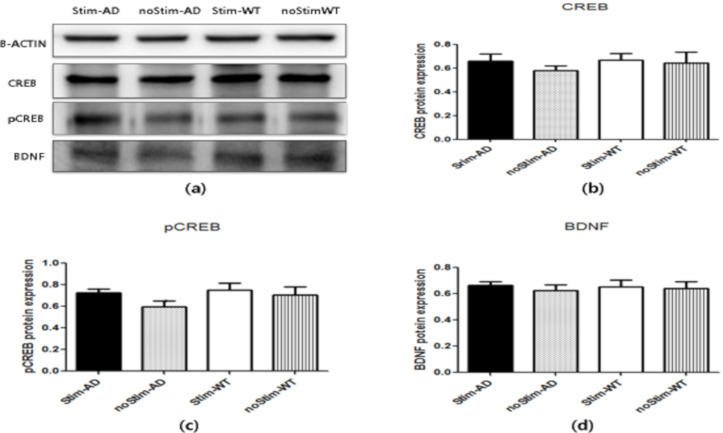
Levels of cyclic amp-response element binding protein (CREB), phosphorylated CREB (pCREB), and Brain-derived neurotrophic factor (BDNF) expression. (**a**) Western blot of the hippocampal samples from each group shows the representative bands of CREB, pCREB, and BDNF. (**b**) The bars represent the quantitative CREB expression levels. (**c**) The bars represent the quantitative pCREB expression levels. (**d**) The bars represent the quantitative BDNF expression levels. Data are presented as mean ± SD and we performed a Kruskal–Wallis test and Tukey’s test using ranks as a post hoc analysis.

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
