# Peer review of "Cognitive Function Improvement in Mouse Model of Alzheimer’s Disease Following Transcranial Direct Current Stimulation"

_brainsci, 2020, doi:10.3390/brainsci10080547_

Round 1
Reviewer 1 Report
In this manuscript, Kim et al. studied the effect of tDCS on hippocampal LTP in a mouse model of AD (5xFAD). The authors found that in hippocampal slices, the decrease in EPSP in 5xFAD model can be rescued by tDCS. Mechanistically, they found a rescued level of pCREB. However, the following issued must be addressed before publication:
- Throughout the manuscript, the authors should tone down claims about the non-significant difference in CREB and BDNF. This includes "CREB and BDNF levels did increase" in the abstract, "CREB protein expression level was higher" (Line 145), "BDNF levels did increase" in Discussion (Line 180), and other places.
- Multiple testing should be addressed.
- Exact p-values should be reported for both figures (Lines 138 and 147).
- Name of statistical test should be mentioned again in figure legends.
- Subheadings in Results (3.1 and 3.2) should be more detailed and descriptive.
- The authors might consider mentioned the difference between mouse models (3xTg versus 5xFAD) when discussing the potential reason for discrepancy with published results (Line 191).
- Typo "Ttranscranial" (Line 53).
Reviewer 2 Report
Kim and colleagues investigate mechanisms underlying transcranial stimulation-induced cognitive improvement in Alzheimer's disease. They found that, by using 5xFAD mouse model, transcranial direct current stimulation enhances LTP in the AD mice with no change in control mice. Furthermore, they detected molecular mechanisms and characterized that the reduced expression level of pCREB is reversed by transcranial direct current stimulation. Overall, the research topic is of great interest in the AD field. However, experimental design and data presentation could be improved.
- Clarify wild-type mice. Were they littermate of 5xFAD mice?
- Clarify why mice were individually housed.
- What is the age of mice used in the each experiment.
- Behavioral tests of learning and memory are necessary to confirm the models and the treatments are comparable to clinical observation.
- Western blot band for pCREB are not clear.
- Is LTP statistically different between control and AD mice?
Round 2
Reviewer 2 Report
The authors answered all my questions.